# Dynamic Inverse Reinforcement Learning for Characterizing Animal Behavior

Zoe C. Ashwood[1,2,*]    Aditi Jha[1,3,*]    Jonathan W. Pillow[1]

[1]Princeton Neuroscience Institute, Princeton University
[2]Dept. of Computer Science, Princeton University
[3]Dept. of Electrical and Computer Engineering, Princeton University
{zashwood, aditijha, pillow}@princeton.edu

## Abstract

Understanding decision-making is a core objective in both neuroscience and psychology, and computational models have often been helpful in the pursuit of this goal. While many models have been developed for characterizing behavior in binary decision-making and bandit tasks, comparatively little work has focused on animal decision-making in more complex tasks, such as navigation through a maze. Inverse reinforcement learning (IRL) is a promising approach for understanding such behavior, as it aims to infer the unknown reward function of an agent from its observed trajectories through state space. However, IRL has yet to be widely applied in neuroscience. One potential reason for this is that existing IRL frameworks assume that an agent's reward function is fixed over time. To address this shortcoming, we introduce *dynamic inverse reinforcement learning* (DIRL), a novel IRL framework that allows for time-varying intrinsic rewards. Our method parametrizes the unknown reward function as a time-varying linear combination of spatial reward maps (which we refer to as "goal maps"). We develop an efficient inference method for recovering this dynamic reward function from behavioral data. We demonstrate DIRL in simulated experiments and then apply it to a dataset of mice exploring a labyrinth. Our method returns interpretable reward functions for two separate cohorts of mice, and provides a novel characterization of exploratory behavior. We expect DIRL to have broad applicability in neuroscience, and to facilitate the design of biologically-inspired reward functions for training artificial agents.

## 1   Introduction

Characterizing the decision-making behavior of humans and animals is a central goal in neuroscience and psychology [1, 2]. Decision-making tasks such as Two-Alternative Forced Choice (2AFC) and bandit problems have been widely studied [3–7], and previous work has developed a variety of models for behavior in these tasks [1, 7–9]. The classic psychometric curve represents one such model [10], and more recent work has focused on models based on reinforcement learning [1, 8, 11, 12]. Such models allow us to understand and compare the decision-making strategies used by humans and animals, and can also provide a low-dimensional description of behavior that can be regressed against neural data [1, 13].

Although a large literature has focused on models of decision-making in simple 2AFC and bandit tasks, comparatively few papers have sought to model behavior in larger, complex natural environments.

---

*These authors contributed equally to this work.

36th Conference on Neural Information Processing Systems (NeurIPS 2022).

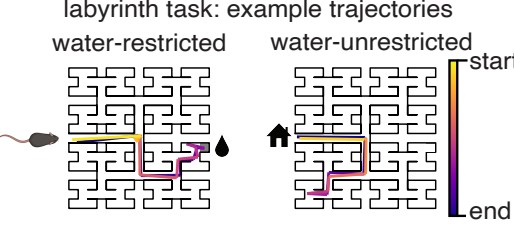

labyrinth task: example trajectories
water-restricted    water-unrestricted

Figure 1: Example mouse trajectories in the labyrinth task of Rosenberg et al. [15]. Water-restricted and unrestricted mice moved freely through a 127 node maze environment for 7 hours. Over the course of the night, each mouse completed over 100 such trajectories, which began at the home port (indicated by an image of a house). For the water-restricted mice, a water port existed at the terminal node in the environment that is shaded grey (next to the water drop image).

[14–16]. In a recent study, Rosenberg et al. [15] introduced a novel experimental paradigm involving water-restricted mice navigating a 127-node labyrinth equipped with a water port at one terminal node (Fig. 1). At each node, the mouse could make up to 4 distinct decisions ('stay', 'go left', 'go right', or 'reverse'). Navigation through a maze is a perfect example of complex yet natural decision-making behavior that remains poorly understood. Although reinforcement learning may seem like the natural framework for modeling such goal-driven behavior, the rewards experienced by these mice are not obvious to the experimenter. Indeed, as noted in Rosenberg et al. [15], the observed trajectories indicate that mice are not only motivated by the extrinsic water reward, but also by intrinsic rewards such as their curiosity to explore the environment.

Inverse reinforcement learning (IRL) [17–20] addresses the problem of inferring the unknown reward function of an agent. Given access to the agent's trajectories as it interacts with the environment, IRL identifies the states and actions that the agent finds rewarding. While IRL has found many successes in robotics [21, 22] and healthcare contexts [23–25], existing IRL methods have not found broad application in neuroscience. One plausible reason for this dearth is that IRL methods typically assume that the unknown reward function is fixed over the course of the agent's trajectory. Yet in many real-world decision-making tasks, rewards can change over time. For example: the goals of a mouse, and consequently its intrinsic rewards, can change with time depending on factors such as fatigue, satiation and curiosity. This motivated us to develop an IRL method for characterizing animal decision-making in complex environments such as the labyrinth, in which rewards may vary over time.

Here we propose dynamic inverse reinforcement learning (DIRL), an IRL method that allows for time-varying reward functions. DIRL parametrizes the animal's reward function as a time-varying linear combination of a small number of spatial reward maps, which we refer to as "goal maps". A single goal map is a function specifying the amount of reward at each state in the environment. (Classic IRL methods thus seek to identify a single, fixed goal map.) DIRL posits the existence of multiple goal maps with time-varying weights, which allows the instantaneous reward function to vary in time. For example, a "water" goal map could have high levels of reward at the water port, while an "explore" reward map might have reward evenly dispersed throughout the environment. Time-varying weights then modulate the extent to which each goal map is active at any timestep. We introduce an inference method, which extends a fixed reward IRL framework by Ziebart et al. [20], to allow us to obtain both the goal maps and the time-varying weights from state-trajectory data. Finally, we demonstrate the application of our framework in simulation (gridworld and labyrinth environments), as well as on real mouse decision-making data from Rosenberg et al. [15]. Our method recovers interpretable reward functions for both cohorts of mice studied there, and reveals an "explore" goal map for one of the cohorts. While exploration remains poorly understood in neuroscience [26], our method offers a powerful framework for characterizing exploratory, as well as exploitative behavior from an animal's trajectories alone.

## 2   Related Work

The neuroscientific literature describing models of decision-making is vast, including both normative [1, 8] and descriptive models [9, 10, 27–30]. While some of this work considers models of decision-making that vary over time, most such models have focused on 2AFC or bandit tasks, and do not scale easily to complex decision-making tasks such as navigation. To the best of our knowledge, IRL has found relatively few applications in neuroscience. One exception is Schultheis et al. [31]

which performed inverse optimal control to infer the cost function associated with sensorimotor behavior. Their targeted application—modeling the cost function optimized by a human performing a reaching task—is very different to ours, which results in different assumptions in their work. Another exception is Yamaguchi et al. [32] which described an IRL framework for identifying the thermotactic strategies used by C. elegans. Relative to the work we present here, Yamaguchi et al. [32] assumed a fixed reward function, and was restricted to the case of a linearly-solvable MDP. In Kwon et al. [33], the authors considered the problem of "Inverse Rational Control" and inferred the parameters governing the evolution of an animal's belief state and subjective reward function within a POMDP (partially observed MDP) framework. Finally, while not the main focus of their paper, Reddy [16] used the static IRL framework of Ziebart et al. [19] to infer the rewards optimized by mice navigating in the Rosenberg et al. [15] task.

The literature on IRL [17–20, 34–37] is extensive, but most studies have assumed that the true underlying reward function is fixed over time. Of particular relevance to our work is the Maximum Causal Entropy (MCE) framework of Ziebart et al. [20]. MCE infers a reward function under the assumption that an agent seeks to maximize both discounted future reward and the discounted future entropy of its policy. We also use discounted future entropy to regularize the goal maps in DIRL, making our method an extension of MCE to the case of time-varying rewards.

Finally, the IRL frameworks of Babes-Vroman et al. [38], Choi and Kim [39] and Likmeta et al. [40] pursued a related aim to ours, allowing for multiple agents with differing reward functions. However, these frameworks did not allow for single agents with time-varying rewards. Surana and Srivastava [41] developed a Bayesian non-parametric method that assumed that an agent's trajectory could be partitioned into distinct behavioral states, where each discrete state had a unique reward function; here, we instead consider the case where rewards vary continuously over time.

## 3 DIRL: Dynamic Inverse Reinforcement Learning

### 3.1 The Inverse Reinforcement Learning Problem

Let us consider a Markov Decision Process (MDP), $\mathcal{M} = \{\mathcal{S}, \mathcal{A}, \mathcal{T}, r, \gamma\}$, where $\mathcal{S}$ is the state space, $\mathcal{A}$ is the action space, $\mathcal{T} : \mathcal{S} \times \mathcal{S} \times \mathcal{A} \to [0, 1]$ represents the probability of transitioning between states when a certain action is taken, $r : \mathcal{S} \times \mathcal{A} \to \mathbb{R}$ is the reward function, specifying the reward obtained from taking action $a \in \mathcal{A}$ in state $s \in \mathcal{S}$, and $\gamma \in [0, 1]$ is the discount factor. Inverse reinforcement learning [17–19, 34, 38, 39] aims to infer the unknown reward function $r(s, a)$ when given access to $\{\mathcal{S}, \mathcal{A}, \mathcal{T}, \gamma\}$ and $N$ trajectories of agents navigating in this environment, $\mathcal{D} = \{\zeta_1, \zeta_2, ..., \zeta_N\}$. Each trajectory is a sequence of state-action pairs, $\zeta_i = \{(s_1, a_1), (s_2, a_2), ...\}$. Typically, IRL frameworks assume that the reward function does not vary over the course of an agent's trajectory. As discussed earlier, this is a severe limitation when applying IRL in neuroscientific settings, where the subjective value of the rewards an animal receives may vary as a function of time, satiety, thirst, fatigue, curiosity, etc. In the following, we introduce an extension of IRL that allows for time-varying reward functions.

### 3.2 DIRL Generative Model: Goal Maps and Time-Varying Weights

Here we extend the MDP defined above by assuming a time-varying reward function: $r_t(a, s)$. Our goal is to develop methods for inferring this reward function from a set of observed trajectories $\mathcal{D} = \{\zeta_i\}_{i=1}^N$. Following previous work [17–19, 34, 38, 39], we assume that the reward function depends only on the state $s$, allowing us to write the dynamic reward function as $r_t(s)$. However, inferring the reward for every state at every timepoint still requires learning $|\mathcal{S}| \times T$ parameters. We therefore make two assumptions to reduce the number of parameters and make inference tractable: (1) we model the reward function as low rank, parameterized by a small number of goal maps modulated by a set of time-varying weights; and (2) we impose a prior encouraging these weights to vary slowly in time.

In our approach, the time-varying reward function has the following low-rank representation:

$$r_t(s) = \sum_{k=1}^{K} \alpha_{k,t} u_{k,s}, \tag{1}$$

where $\mathbf{u}_k \in \mathbb{R}^{\mathcal{S}}$ represents the $k$'th goal map, and $\alpha_{k,t} \in \mathbb{R}$ is the weight on this goal map at timestep $t$, and $K$ is the rank of the representation. Each goal map specifies a reward level for each state in

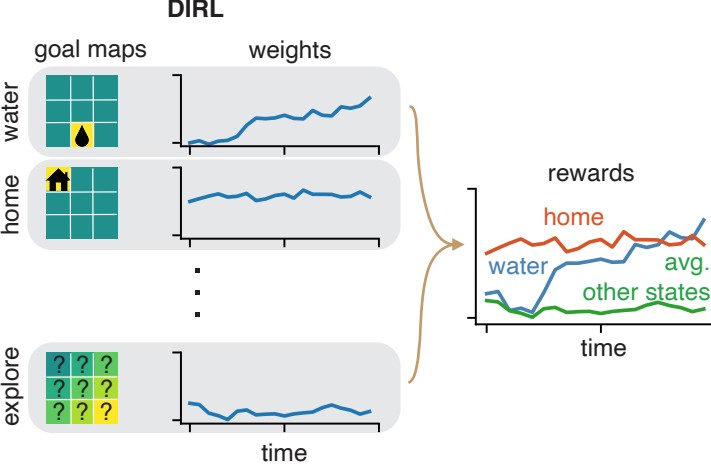

Figure 2: DIRL model schematic. We parametrize the time-varying reward function as a linear combination of a small number of spatial goal maps, where the weights on each map can vary over time. A goal map is a map of the environment, indicating the reward available in each state.

the environment, while weight $\alpha_{k,t}$ specifies the contribution of goal map $k$ to the animal's reward function at $t$.

To impose smoothness over time, we place a Gaussian random walk prior over weight trajectories:

$$\alpha_{k,t} = \alpha_{k,t-1} + \epsilon_k; \qquad \epsilon_k \sim \mathcal{N}(0, \sigma_k^2), \tag{2}$$

where $\sigma_k^2$ is a hyperparameter controlling the variance of the weight changes. This prior reflects our belief that the factors influencing an animal's subjective experience of reward (e.g., thirst, hunger, fatigue) vary slowly relative to the timescale of individual decisions. The low-rank assumption reduces the number of parameters specifying the reward function from $(|\mathcal{S}| \times T)$ to $(|\mathcal{S}| + T)K$, a massive reduction provided $K \ll \min(|\mathcal{S}|, T)$. The smoothness assumption allows us to further reduce the effective number of parameters, as the weight trajectories $\alpha_{k,t}$ become more correlated with decreasing variance $\sigma_k^2$ [42].

Figure 2 illustrates the resulting generative model using a simplified $3 \times 3$ gridworld. In this example, there are multiple goal maps, one with reward located only at the water port, a second with reward located only at the home state, and a third with reward distributed broadly across states, which is associated with exploratory behavior. Each map has a time-varying weight that determines its contribution to the animal's total reward function at a given moment in time. In this example, the "home" goal map dominates the reward function at the beginning of the session, while the "water" goal map dominates at the end of the session (e.g., reflecting the animal becoming thirsty).

To model decision-making behavior, we assume that animals seek to maximize the discounted expected future reward under a maximum entropy policy [19, 20, 37, 43, 44], given by:

$$\pi_t(s, a) = \frac{e^{Q^t(s,a)}}{\sum_{a' \in \mathcal{A}} e^{Q^t(s,a')}} \qquad \forall\, s \in \mathcal{S}, a \in \mathcal{A}, t \in \{1, ...T\}, \tag{3}$$

where $Q^t(s, a)$ is the soft Q-function for state $s$ and action $a$ at time $t$:

$$Q^t(s, a) = r_t(s) + \gamma \sum_{s'} P(s' \mid s, a) \log \left( \sum_{a'} \exp Q^t(s', a') \right), \tag{4}$$

which arises from the reward function by performing soft value iteration [34] (see Supplementary Materials, SM, for further details). This is a common choice of policy in IRL frameworks as it is easily differentiable (unlike maximizing or "greedy" policies), and it has also been widely applied to data from both humans and animals [13, 28]. Note that our formulation does not require a temperature parameter, as this is directly incorporated into the scale of the time-varying weights: larger (smaller) weights give rise to more (less) deterministic policies.

### 3.3 DIRL inference procedure

During inference, our objective is to learn the time-varying weights $\{\boldsymbol{\alpha}_k\}_{k=1}^{K}$, as well as the goal maps $\{\mathbf{u}_k\}_{k=1}^{K}$ from the trajectories of an agent (animal). To do so, we alternately optimize the goal maps and time-varying weights to maximize the log-posterior of the observed trajectories in $\mathcal{D}$ under our model. Let $\mathbf{u} \in \mathbb{R}^{\mathcal{S}K}$ be a vector of concatenated goal maps $\{\mathbf{u}_k\}_{k=1}^{K}$ stacked vertically, and similarly let $\boldsymbol{\alpha} \in \mathbb{R}^{TK}$ contain the concatenated time-varying weights $\{\boldsymbol{\alpha}_k\}_{k=1}^{K}$. stacked vertically. We, first, initialize the parameters randomly, such that the elements of the goal maps are chosen from $U(0,1)$ and the time-varying weights are Gaussian distributed. We then perform coordinate ascent to iteratively update the time-varying weights and the goal maps while holding the other set of parameters constant.

Concretely, we obtain the goal map updates $\mathbf{u}$ by maximizing the following objective using gradient ascent:

$$\mathbf{u}^* = \arg\max_{\mathbf{u}} \sum_{i=1}^{N} \sum_{(s_t, a_t) \sim \zeta_i} \log \pi_t(s_t, a_t) - \lambda ||\mathbf{u}||^2 \tag{5}$$

where $\pi_t$ is the policy given by Eq. 3 and $\lambda ||\mathbf{u}||^2$ represents an L2 regularizer.

We then update the time-varying weights $\boldsymbol{\alpha}$ by first updating the reward function and policy (Eq. 3) with the new goal maps, $r_t(s) = \sum_k \alpha_{t,k} u_{k,s}^*$. We then use gradient ascent to perform the optimization:

$$\boldsymbol{\alpha}^* = \arg\max_{\boldsymbol{\alpha}} \left( \sum_{i=1}^{N} \sum_{(s_t, a_t) \sim \zeta_i} \log \pi_t(s_t, a_t) - \tfrac{1}{2}\log|C| - \tfrac{1}{2}\boldsymbol{\alpha}^\top C^{-1}\boldsymbol{\alpha} \right). \tag{6}$$

The last two terms in this objective correspond to the negative log of the Gaussian prior on $\boldsymbol{\alpha}$ (Eq. 2), where $C^{-1} = D^\top \Sigma^{-1} D$ is the inverse prior covariance, with $D$ a block diagonal matrix of $K$ identical $T \times T$ first-order difference matrices (with 1s on the diagonal and -1s on the sub-diagonal), and $\Sigma^{-1}$ is a diagonal matrix with inverse noise-variances $1/\sigma_k^2$ along the diagonal.

We iteratively update the goal maps and time-varying weights using Eq. 5 and Eq. 6 until convergence. (See Alg. 1 for pseudo-code). We consider the number of goal maps $K$, the discount factor $\gamma$, the strength of the goal map prior $\lambda$, as well as the noise variances associated with the time-varying weights, $\{\sigma_k\}_{k=1}^{K}$, to be hyperparameters. To restrict the number of hyperparameters, we set $\sigma_k = \sigma \; \forall k$. We then swept across a broad range of values for all hyperparameters (see SM for the full list of values considered) and selected the values that optimized the log-likelihood of a set of held-out trajectories.

---

**Algorithm 1:** DIRL Inference Procedure

---

**Input 1:** MDP state and action spaces, transition matrix: $(\mathcal{S}, \mathcal{A}, \mathcal{T})$ ;
**Input 2:** $N$ trajectories, $\mathcal{D} \equiv \{\zeta_i\}_{i=1}^{N}$ ;
**Input 3:** Hyperparameters: no. of goal maps $K$, noise variances $\{\sigma_k\}$, discount factor $\gamma$,
 strength of goal map prior, $\lambda$ ;
**Output:** Parameters governing the rewards $\{\mathbf{u}_k, \boldsymbol{\alpha}_k\}_{k=1}^{K}$, where $\mathbf{u}_k \in \mathbb{R}^{\mathcal{S}}, \boldsymbol{\alpha}_k \in \mathbb{R}^{T}$ ;

Let $\mathbf{u} = [\mathbf{u}_1, ...\mathbf{u}_k], \boldsymbol{\alpha} = [\boldsymbol{\alpha}_1, ...\boldsymbol{\alpha}_k]$ ;
Initialize $\mathbf{u}^0, \boldsymbol{\alpha}^0$ ;
**for** $iter = 1...N_{iter}$ **do**
    Calculate rewards $r_t(s) = \sum_k \alpha_{k,t}^{\text{iter}} u_{k,s}^{\text{iter}} \; \forall s \in \mathcal{S}, t \in \{1...T\}$;
    Get policy using soft value iteration : $\pi_t(s, a) = \dfrac{e^{Q^t(s,a)}}{\sum_{a'} e^{Q^t(s,a')}} \; \forall(s, a, t)$;
    Update $\mathbf{u}^{\text{iter}+1}$ by maximizing the log-posterior of trajectories (Eq. 5).;
    Update rewards $r_t(s) = \sum_k \alpha_{k,t}^{\text{iter}} u_{k,s}^{\text{iter}+1}$ and learn new policy $\pi_t(s, a)$;
    Update $\boldsymbol{\alpha}^{\text{iter}+1}$ by maximizing the log-posterior, with noise variances $\{\sigma_k\}$ (Eq. 6).;
**end**
Output $\mathbf{u}^{N_{\text{iter}}}, \boldsymbol{\alpha}^{N_{\text{iter}}}$, ;

---

# 4 Results

## 4.1 Application to a simulated gridworld environment

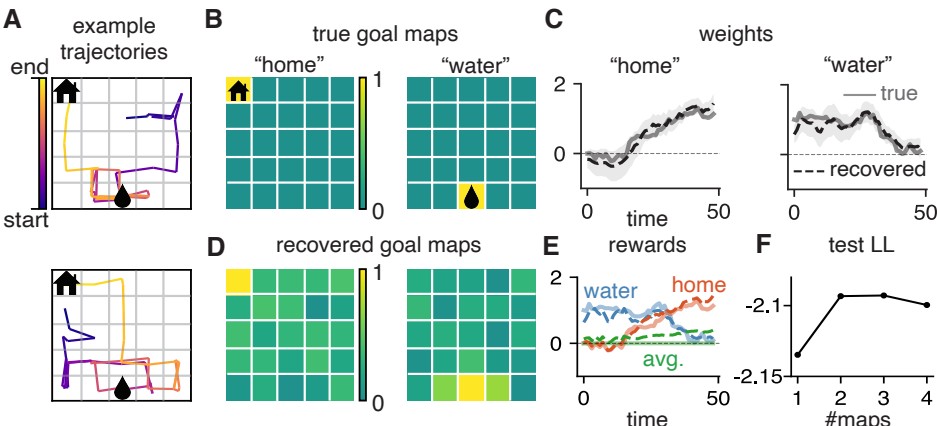

Figure 3: Simulations on a $5 \times 5$ gridworld. **(A)** Example expert trajectories when the time-varying reward function is obtained using the generative goal maps shown in B and the time-varying weights shown in C. **(B)** Generative goal maps: the first map has a high reward at the "home state" (upper left), the other has a high reward at the the "water state" (bottom center). **(C)** Time varying weights for the home and water goal maps: solid lines show the generative parameters, while dotted lines show the recovered parameters along with a 95% confidence interval. Error bars are computed via the inverse Hessian of the log-posterior of Eq. 6 at the MAP estimate of the weights. **(D)** Recovered goal maps. **(E)** Rewards for the home and water states are shown in red and blue respectively. The average reward for the remaining states is shown in green. Solid lines show the generative rewards, while the dotted lines show the inferred rewards. **(F)** Held-out test set performance as a function of the number of goal maps. Higher values are better; units are bits per decision.

We first demonstrate our method on simulated trajectories in a $5 \times 5$ gridworld environment, with 5 actions per state (up, down, left, right, stay). We generated two goal maps for this environment: a "home" map and a "water" map, which were rewarding only at the home state and the water state, respectively (Fig 3B). Corresponding to these goal maps, we also generated time-varying weights (Fig. 3C, solid lines) for 50 timesteps with the random-walk prior of Eq. 2 (for $\sigma_k = 2^{-3.5}$; this was chosen to provide adequate variation in the reward function during the time period considered). The weight for the water map started high but decreased over time, thus making the water state the most rewarding for the first $\sim$25 timesteps (Fig. 3E, blue solid line). In contrast, the weight on the home map started small but increased so that the home state became the most rewarding state at the end of the 50 timesteps (Fig. 3E, red solid line). All of the other states in the environment had a constant reward of 0. In order to generate trajectories corresponding to this reward function (Fig. 3E), we used soft value iteration to learn the corresponding optimal time-varying policy (Eq. 3). We then executed this policy in order to obtain 200 trajectories (a similar number to the number of trajectories we have for the real dataset discussed later), two examples of which are shown in Fig. 3A.

Next, we applied our IRL inference method so as to learn the goal maps and time-varying weights from 80% of the generated trajectories. Fig. 3F shows the log-likelihood of the remaining 20% of trajectories as a function of the number of goal maps. The held-out test log-likelihood is equally high for 2 and 3 maps, so we focus on the 2-map solution in order to be able to compare with the generative parameters. It is important to note that we can only recover rewards at each timestep up to an additive constant, as the policy remains unchanged upon the addition of a constant to the rewards. Further, scaling of the goal maps accompanied by an inverse scaling of the time-varying weights also leaves the recovered rewards unchanged. Thus, to compare to the generative parameters, we perform a post hoc processing method to the recovered parameters to handle all such invariances (details in SM). Figures 3C and D show that our method allows us to accurately recover the generative goal maps and time-varying weights from the simulated trajectories. Finally, combining the goal maps and time-varying weights, we are able to accurately match the generative time-varying rewards for

different states in the gridworld (Fig. 3E). We also simulated trajectories in a 127-node labyrinth environment (akin to [15], Fig 1), and confirmed that we were able to recover goal maps and time-varying weights (see SM for details). We focus on the gridworld simulations here to demonstrate the versatility of our approach across environments.

## 4.2 Application of DIRL to real mouse trajectories

Next, we applied our framework to the trajectories of real mice navigating (in the dark) in a 127-node labyrinth environment [15]. In this task, two cohorts of 10 mice moved freely through the labyrinth over the course of 7 hours. The first cohort was water-restricted and the mice were provided with a water port at one of the terminal nodes (shown in gray in Fig. 1). The second cohort was not water-restricted and did not have access to the water port. We show an example trajectory for an animal in each cohort in Fig. 1. Over the course of the night, each animal completed over a hundred such trajectories, with some animals completing many more.

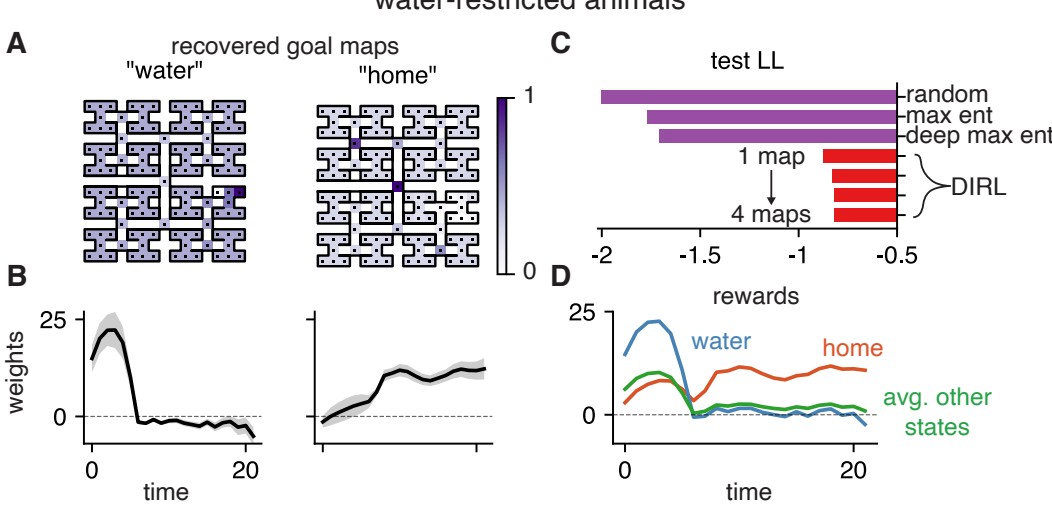

Figure 4: Inferred time-varying rewards for water-restricted mice. (**A**) Inferred goal maps for the water-restricted mice: a "water" map and a "home" map. Dots indicate each of the 127 nodes in the labyrinth environment. (**B**) Recovered time-varying weights for the same animals. Each timestep corresponds to 1 second. (**C**) Model comparison: comparison of DIRL on held-out trajectories to a random policy, as well as the Maximum Entropy IRL framework of [19] and the Deep Maximum Entropy IRL framework of [34]. Higher test log-likelihood is better; test log-likelihood has units of bits/decision. (**D**) Inferred time-varying rewards for the home state, the water state, and the average reward for the remaining states.

### 4.2.1 Inferring interpretable reward functions from water-restricted mice

We began our investigation by applying our method to a subset of trajectories for the water-restricted animals. Here, we anticipated being able to identify the water port as highly rewarding (in contrast to the water-unrestricted animals). Due to the high variability in trajectories across animals and over the course of the 7 hours, and to be able to obtain trajectories corresponding to similar goal maps and with a similar time course, we used an unsupervised clustering algorithm (based on dbscan [45] and using the Levenshtein distance metric; full details are in the SM) to identify similar trajectories. This procedure had the advantageous side effect of excluding the first 25 or so trajectories for each animal: our focus is not on characterizing learning. Overall, we obtained 200 trajectories for this cohort. Each timestep in a trajectory was recorded at a one second interval.

We then fit the goal maps and time-varying weights to 160 of these trajectories, and held out the remaining 20% of trajectories as a validation set. We found that validation log-likelihood began to level off at two maps (Fig. 4C), so we focus on the 2 map solution here. The 3 map solution is

shown in the SM (where the recovered water goal map is simply repeated). Overall, we found that we were, indeed, able to recover a "water" goal map (Fig 4A), with a large reward at the water state and small rewards elsewhere, as well as a "home" goal map with a large reward at the home state. The recovered time-varying varying weights (Fig 4B) for the water map were high at the beginning – the mouse likely entered the labyrinth due to being thirsty – and later on, as satiation and fatigue set in, tailed off. In contrast, the weights corresponding to the home map started low, but increased over time. The final reward function (Fig 4D) reflects the same dynamics: the water state was highly rewarding for the mouse at the start of its trajectory, while the home state became the more rewarding state after ∼6 seconds. The other states in the labyrinth offered only a small intrinsic reward for the mouse throughout the course of its trajectory.

### 4.2.2 'Exploratory' maps inferred from water-unrestricted mice

We then moved on to examining the trajectories of the water-unrestricted cohort. Relative to the water-restricted mice, the goal maps for these mice were not obvious a priori – there were no extrinsic reward locations for this cohort in the labyrinth. We began by applying the same clustering algorithm as for the water-restricted animals (discussed in the SM) and identified 207 trajectories for our analysis.

We fit the time-varying weights and goal maps to 80% of these trajectories, and held out the remaining 20% as a validation set. As we show in Fig. 5C, the validation log-likelihood is best for the 2 map solution. While we recover (Fig 5A) a "home" map once again, a new map – that was not present for the water-restricted animals – also appears. This map is rewarding at many states throughout the maze, but is very unrewarding at the home port, and is very rewarding at a state that is close to the home port (which we refer to as the "explore" state). Hence, when the animal places a positive weight on this map, the mouse wants to leave the home state and venture into the maze. For these reasons, we refer to this map as the "explore" map. The time-varying weights (Fig 5B) for the home map increase with time; this captures the tendency of these animals to go back to the home state towards the end of the trajectory. The weights corresponding to the explore map are positive at the beginning of the trajectory, capturing the exploratory behavior of these animals when they enter the labyrinth. Finally, the inferred rewards (Fig 5D) reveal that, as we would expect for this cohort, the time-varying reward for the water port is almost identical to the average reward function (across all other states in the labyrinth). We can also see that the reward for the home state rises much more rapidly over time than the reward for other states.

### 4.2.3 DIRL outperforms existing IRL approaches

Finally, we compare the performance of our method with two popular IRL frameworks: the maximum entropy IRL framework of Ziebart et al. [19] and the deep maximum entropy IRL framework of Wulfmeier et al. [34]. In comparison to DIRL, these methods learn a static reward function. We use an open-source implementation of these frameworks [46] and infer the reward functions for each of these methods for the two cohorts of mice studied above. Using the retrieved reward functions, we then obtain the corresponding optimal policy and use it to compute the log-likelihood of the validation set of trajectories. In Figures 4C and 5C, we demonstrate that our method dramatically outperforms these existing methods at explaining the held-out trajectories of both water-restricted and unrestricted mouse cohorts. In the SM, we also show that the trajectories generated by DIRL very well resemble the behavior of mice in the labyrinth, as compared to those generated from [19] and [34].

## 5 Discussion

In this work we develop DIRL, a novel inverse reinforcement learning framework for characterizing the behavior of animals during complex decision-making tasks. Our framework infers the time-varying reward functions of animals from their trajectories alone. We validated our framework on simulated data in a gridworld environment, and applied it to two cohorts of 10 mice navigating in a labyrinth [15]. Our method provided distinct and interpretable reward functions for both cohorts: the water-restricted mice assigned a high reward to the water state upon entering the labyrinth, while the water-unrestricted mice were motivated to leave home and explore at the beginning of their trajectories. As time passed and the mice became fatigued, the reward for the home state increased for animals in both cohorts. Our method dramatically outperfomed IRL methods with fixed reward

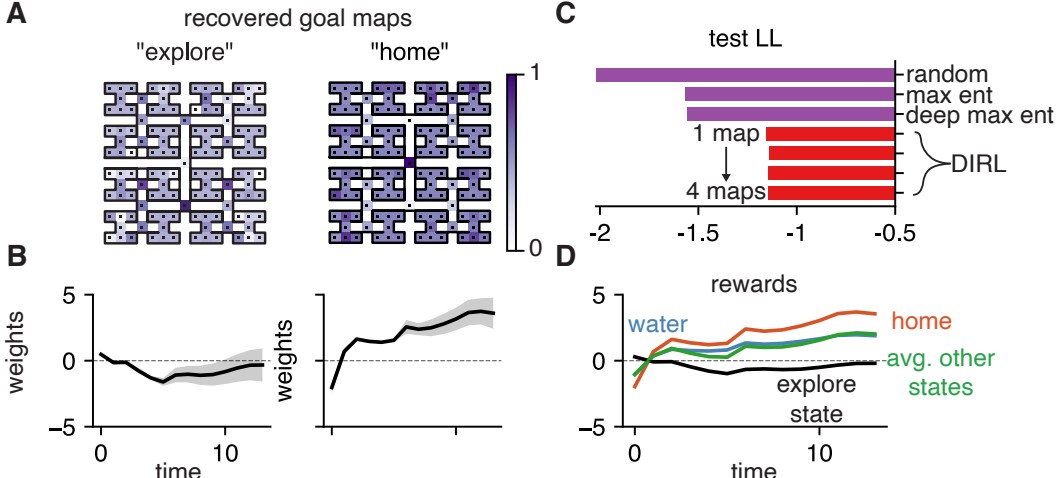

Figure 5: Inferred time-varying rewards for water-unrestricted mice. **(A)** Inferred goal maps for this cohort: an "explore" map and a "home" map are returned. Dots indicate each of the 127 nodes in the labyrinth environment. **(B)** Recovered time-varying weights for the same animals. **(C)** Model comparison: comparison of DIRL on held-out trajectories to a random policy, as well as the Maximum Entropy IRL framework of [19] and the Deep Maximum Entropy IRL framework of [34]. Higher test log-likelihood is better; test log-likelihood has units of bits/decision. **(D)** Inferred time-varying rewards for the home state, the water port, the 'explore' state (the highly rewarding state in the 'explore' goal map that is closest to home) and the average reward for the remaining states. Here the average reward line is almost on top of the line corresponding to the water port.

functions, as quantified by the log-likelihood of held-out trajectories, indicating a clear need for tailored IRL approaches for neuroscientific applications. Finally, our method is computationally efficient and infers the time-varying reward function from ∼4000 decisions in the 127-node labyrinth environment in 20 minutes on a laptop.

One exciting finding of our work is the discovery of an "explore" goal map for the water-unrestricted mouse cohort. In general, exploration in animals is not well understood [26]. DIRL offers a novel unsupervised approach for characterizing exploration from behavior alone. Future work could build upon the framework we present here in order to provide normative explanations for the nature of the explore map. DIRL could also be used to explore the neural underpinnings of exploration, for example by correlating fluctuations in the inferred time-varying rewards with recorded neural activity. Finally, while we developed DIRL with neuroscientific applications in mind, our framework is general and could be applied in other scientific fields. For example, DIRL could be applicable in healthcare settings, such as those considered in Hüyük et al. [47]. In Hüyük et al. [47], the authors acknowledge that the policies used by healthcare providers to allocate organ donations have varied over time. By applying DIRL to such a dataset, it may be possible to recover the latent goals of policy-makers or medical practitioners when deciding how best to allocate organs to patients.

In addition to the above-mentioned scientific applications, we believe that DIRL also provides future directions for the development of RL algorithms. Relative to animals, it is often challenging to get artificial agents to explore in sparse reward environments [48]. With access to the internal reward function that motivates mice, we hope that our framework may be useful for inspiring better reward functions for training artificial agents to navigate in analogous environments (such as in [49] where artificial agents navigated in an analogous depth-6 binary tree environment). Another valuable future direction could involve developing alternative ways of formulating the problem of IRL with time-varying goals, such as casting our problem as a Hidden-Mode Hidden Markov Model [50] —a type of POMDP— and then performing IRL with POMDPs [51].

We will close by discussing some limitations of our work. Firstly, our framework requires over a hundred trajectories to infer an animal's time-varying reward function. While it may be possible to

reduce this number (with, for example, a careful choice of prior), it is easy to conceive of failure modes where a single decision reveals nothing about the active goal map. In practice, having access to multiple trajectories (or several decisions) can significantly reduce the uncertainty in the recovered time-varying weights and goal maps. Next, our approach may not scale well to high-dimensional state spaces, as we currently learn a separate reward for each state in each goal map. We don't anticipate this being a problem in the neuroscientific applications that we discuss in this work (where the state-space is 127 dimensional) but an extension of our framework could involve learning a low-dimensional representation of each state in the environment via a deep network, and then learning time-varying rewards in this embedding space. This would allow us to scale our approach to higher dimensional state spaces, while also allowing for generalization across states. Finally, our inferred rewards rely on the animal's policy being the Boltzmann policy of Eq. 3. However, assuming this form for the policy is not atypical in applications of reinforcement learning to human or animal decision-making data [13, 28]. Overall, we believe that the advantages of using our method outweigh its limitations, and that we present a new, flexible framework for characterizing animal behavior in complex environments.

## Acknowledgments and Disclosure of Funding

We thank Benjamin Cowley, Nathaniel Daw, Rachit Dubey and Karthik Narasimhan for useful comments and discussions at various points during this project. We thank the anonymous NeurIPS reviewers for their insightful feedback and helpful suggestions for improvement. The mouse in Figure 1 was provided by BioRender. This work was supported by grants from the Simons Collaboration on the Global Brain (SCGB AWD543027), the NIH BRAIN initiative (NS104899 and R01EB026946), and a U19 NIH-NINDS BRAIN Initiative Award (5U19NS104648).

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
