# Supplementary material (SM)

# Contents

## A    Additional model training and hyperparameter selection details

### A.1    Training details for simulations in the gridworld environment

For inferring the time-varying rewards in the gridworld environment, we varied the number of maps, $K$, between 1 and 4 (shown in Fig. 3F), and varied the learning rates for the weights and the goal maps to take on values in $[0.05, 0.01, 0.005, 0.001]$. We fixed the discount factor $\gamma$, and the noise variance of the random walk prior $\sigma$ to their generative values of $0.9$ and $2^{-3.5}$ respectively. Finally, we initialized with 10 different random seeds. We selected hyperparameters using validation-set log-likelihood; the learning rates that resulted in the fits shown in Fig. 3 were 0.05 for the weights and 0.001 for the maps.

### A.2    Method to align generative and recovered parameters

As mentioned in section 4.1, we perform a post hoc processing method to the parameters recovered by DIRL in order to compare them to the generative parameters. This is because of the fact that several parameter settings can result in the same time-varying policy (Eq. 3). Firstly, the recovered rewards are only identifiable up to an additive constant, as the policy (Eq. 3) does not change upon the addition of a constant to the rewards at a fixed time point. Secondly, our reward parameterization (Eq. 1) is invariant to any scaling of the goal maps accompanied by an inverse scaling of the time-varying weights. Finally, the retrieved maps and weights can be permuted relative to the generative maps and weights (e.g. generative goal map 1 becomes recovered goal map 3). Keeping the above invariances in mind, we come up with a procedure to align the generative and recovered goal maps and time-varying weights.

Concretely, let the superscript $g$ and $d$ denote the generative and the DIRL-recovered parameters respectively, such that the generative and recovered rewards for state $s$ at time point $t$ are given as:

$$r_t^g(s) = \sum_k \alpha_{k,t}^g u_{k,s}^g \tag{S1}$$

$$r_t^d(s) = \sum_k \alpha_{k,t}^d u_{k,s}^d \tag{S2}$$

Here, $\alpha_{k,t}$ corresponds to the weight at time $t$ on the $k$th goal map, $u_k \in \mathbb{R}^{\mathcal{S}}$. First of all, our recovered parameters may not be in the same order as the generative parameters: $u_k^d$ may correspond to $u_{k'}^g$, where $k \neq k'$. Hence, we apply our alignment algorithm on all possible permutations of the $K$ recovered maps and weights. We select the permutation that leads to the smallest $l_2$ distance between the aligned generative and recovered parameters.

Now, for each permutation of the recovered parameters, to align them with the generative parameters, we perform a sign conversion. Our reward parameterization allows for sign invariances such that:

$$\alpha_{k,t} u_{k,s} = (-\alpha_{k,t})(-u_{k,s}) \tag{S3}$$

This means that $\{\alpha_k^d, u_k^d\}$ can both have signs that are flipped respective to the corresponding generative parameters, $\{\alpha_k^g, u_k^g\}$. We use the time-varying weights to determine whether or not to flip the signs of the recovered parameters, $\{\alpha_k^d, u_k^d\}$. In particular, we flip the signs of the recovered parameters if

$$\Big(\sum_t |\alpha_{k,t}^g - \alpha_{k,t}^d|^2\Big)^{1/2} > \Big(\sum_t |\alpha_{k,t}^g + \alpha_{k,t}^d|^2\Big)^{1/2}. \tag{S4}$$

We have two remaining free degrees of freedom to set: (1) the constant, state-independent offset, $c_k$, that can be added to each goal map without affecting the recovered time-varying policy and (2) the scale of the goal maps, $s_k$, which, accompanied by an inverse-scaling of the time-varying weights, has no effect on the recovered reward function. We set $c_k$ and $s_k$ so that the maximum and minimum values of the recovered goal maps match those of the generative goal maps. Thus, $\{c_k, s_k\}$ are solutions of the following set of equations:

$$\min_{s \in \mathcal{S}} s_k u_{k,s}^d + c_k = \min_{s \in \mathcal{S}} u_{k,s}^g \tag{S5}$$

$$\max_{s \in \mathcal{S}} s_k u_{k,s}^d + c_k = \max_{s \in \mathcal{S}} u_{k,s}^g \tag{S6}$$

Since we performed a sign flipping operation earlier, we can assume, without loss of generality, that $s_k > 0 \; \forall k$. These assumptions allow us to solve the above set of equations analytically, leading to closed-form expressions for $s_k$ and $c_k$. Let's define $m_{0,k}^d \equiv \min_s u_{k,s}^d$, $m_{1,k}^d \equiv \max_s u_{k,s}^d$ and $m_{0,k}^g \equiv \min_s u_{k,s}^g$, $m_{1,k}^g \equiv \max_s u_{k,s}^g$. Then:

$$c_k = \frac{m_{0,k}^g m_{1,k}^d - m_{0,k}^d m_{1,k}^g}{m_{1,k}^d - m_{0,k}^d} \tag{S7}$$

$$s_k = \frac{m_{1,k}^g - m_{0,k}^g}{m_{1,k}^d - m_{0,k}^d} \tag{S8}$$

Hence, we obtain our aligned goal maps and time varying weights as follows:

$$u_{k,s}^{d'} = s_k u_{k,s}^d + c_k \; \forall s \in \mathcal{S}, k \in \{1, ...K\} \tag{S9}$$

$$\alpha_{k,t}^{d'} = \frac{\alpha_{k,t}^d}{s_k} \; \forall t \in \{1, ...T\}, k \in \{1, ...K\} \tag{S10}$$

As mentioned earlier, we apply the above set of transformations (sign-flipping, scaling and applying offsets) for each possible permutation of the recovered parameters, and choose the permutation that results in the minimum $l_2$ distance between the recovered and generative parameters.

## A.3 Training details for experiment on the mouse dataset

To fit DIRL on the trajectories obtained from the cohorts of water-restricted and water-unrestricted mice, we swept over the range of hyperparameters mentioned in Table. S1. For each hyperparameter setting, we initialized DIRL with

| Hyperparameter | Range |
|---|---|
| no. of maps, $K$ | $\{1, 2, 3, 4\}$ |
| noise variance, $\sigma$ | $[2^{-3.5}, 1]$ |
| discount factor, $\gamma$ | $\{0.99, 0.9, 0.7\}$ |
| $l_2$ coefficient, $\lambda$ | $\{0, 1e-3, 1e-2, 1e-1, 1.0\}$ |
| learning rate for maps | $\{0.05, 0.01, 0.005]\}$ |
| learning rate for weights | $\{0.05, 0.01, 0.005\}$ |

Table S1: Hyperparameters for fitting DIRL on the mouse dataset

multiple different seeds (between 2 and 10, depending on the specific experiment). We selected the best hyperparameters based on validation log-likelihood, as well as interpretability of the retrieved parameters. For example, in Figure 4, we focused on the 2-map solution as opposed to the 3-map solution. Both solutions had comparable validation-set performance, but the 3-map solution corresponded to splitting the 'water' goal map/weights into two slightly different 'water' goal maps: see section D.2.

The results for water-restricted mice (shown in Fig. 4) correspond to 2 maps, $\gamma = 0.7$, $\sigma = 0.25$, $\lambda = 0.001$ and had the learning rate for maps and weights set to 0.005 and 0.05 respectively. The results for water-unrestricted mice (shown in Fig. 5) correspond to 2 maps, $\gamma = 0.99$, $\sigma = 0.09$, $\lambda = 0.001$ and had the learning rates for the maps and weights set to 0.01 and 0.001 respectively.

### A.4 Hyperparameter selection for Max. Ent. and Deep Max. Ent. comparisons

In Figures 4 and 5, we compare the performance of DIRL with two popular IRL frameworks: the maximum entropy framework of Ziebart et al. [1] and the deep maximum entropy framework of Wulfmeier et al. [2]. In order to conduct these comparisons, we used the implementations of these frameworks provided by Lu [3]. For the maximum entropy framework, we varied the discount factor $\gamma$ in $[0.99, 0.9, 0.7, 0.5]$, and varied the learning rate in $[0.001, 0.005, 0.01, 0.05, 0.1]$. For the water-restricted mice shown in Fig. 4, we found that the best-fitting hyperparameters were $\gamma = 0.7$ and the optimal learning rate was 0.1. For the water unrestricted mice, we found that the best-fitting hyperparameters were $\gamma = 0.5$ and learning rate 0.05.

| Hyperparameter | Range |
|---|---|
| discount factor, $\gamma$ | $\{0.99, 0.9, 0.7, 0.5\}$ |
| $l_2$ coefficient, $\lambda$ | $\{0, 1, 10\}$ |
| learning rate | $\{0.001, 0.005, 0.01, 0.05, 0.1\}$ |
| layer width | $\{5, 10, 20, 50\}$ |

Table S2: Hyperparameters for fitting Deep Max. Ent. [2] to mouse data

When fitting the deep maximum entropy framework to the mouse data, we modified the implementation provided in [3] to use 3 fully-connected layers as opposed to 2 fully-connected layers (we anticipated greater depth being better at fitting this data) to map the state identity to the reward for that state. We also allowed for the width of each layer to take on the values shown in Table S2. The implementation provided by [3] uses Exponential Linear Unit (ELU) activation functions, and applies an l2 penalty to the weights of the network. Along with the layer width, we allowed the discount factor, $\gamma$, the strength of the l2 penalty term, $\lambda$, and the learning rate to take on the values specified in Table S2. We found that the optimal hyperparameters for the water-restricted mice were $\gamma = 0.5$, learning rate= 0.05, layer width= 50 and $\lambda = 1$. For the water-unrestricted mice, the optimal hyperparameters were $\gamma = 0.5$, learning rate= 0.1, layer width= 20 and $\lambda = 0$.

## B  Mouse decision-making data

In Figures 4 and 5, we show the results of applying DIRL to the mouse decision-making data of [4]. We obtained the mouse decision-making data for both cohorts of animals ('water-restricted' and 'water-unrestricted') from `https:`

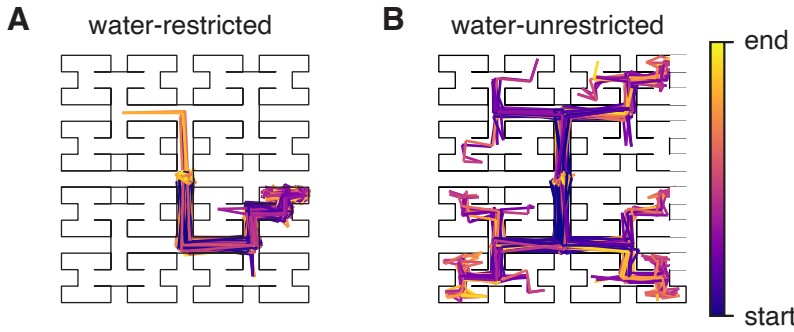

Figure S1: All trajectories overlaid for the water-restricted and unrestricted cohorts studied in text.

In Rosenberg et al. [4], trajectories were obtained by tracking the position of the animal's nose as it moved through the labyrinth environment, and are available at 30Hz. For our analysis, we used the version of trajectories where positions were assigned to the nearest of the 127 maze nodes. So as to reduce the effect of any rounding errors introduced by doing this, we downsampled trajectories so that each timestep in a trajectory corresponded to 1 second. The decision-making data of Rosenberg et al. [4] is naturally segmented into 'bouts', where each bout corresponds to the animal leaving its home cage and ends when the animal returns home. Every trajectory we considered corresponds to a single bout.

## B.1 Clustering algorithm for aligning trajectories across animals and bouts

In Figures 4 and 5, we show the results of applying DIRL to 200 trajectories from the water-restricted animals and 207 trajectories for the water-unrestricted mice, respectively. In order to obtain these trajectories, we applied clustering so as to obtain trajectories corresponding to similar goal maps and similar time-varying weights. Specifically, we represented each trajectory as a string containing the sequence of visited nodes. So that each node was represented by a single character, we first converted each node into its ASCII representation (e.g. node 97 became 'a'). For every pair of trajectories (across animals in the same cohort), we then calculated the Levenshtein distance between the associated strings representing these trajectories. We then used the implementation of dbscan [5] available in scikit-learn [6] (with the maximum distance parameter set to 3 and 6, respectively for the water-restricted and unrestricted animals so as obtain $\sim 200$ trajectories for each cohort, and the minimum number of examples in a cluster set to 5 – we did not optimize these hyperparameters) in order to perform the clustering. We analyzed the trajectories assigned to the biggest cluster for each cohort. Finally, to ensure that all trajectories studied had the same length, we set the desired trajectory length to be the 75th percentile length (we did not optimize this hyperparameter either) across all trajectories in the cluster, and then padded trajectories with lengths shorter than this with the location of the home port (and cut off trajectories longer than this). Given that the water-restricted mice spent 53% of their time in the maze environment at the home cage, while the water-unrestricted mice spent 56% of their time there, this seemed like the natural thing to do. Note: the Levenshtein distance metric naturally selects trajectories of approximately the same length, so the overall amount of padding applied was low.

## B.2 Population summary plot: all trajectories for each cohort

While we show a single trajectory for each cohort in Fig. 1, in Fig. S1 we provide a population summary by showing all trajectories for each cohort overlaid.

# C    DIRL objective function

In standard max-entropy RL (when the rewards are static), the policy is chosen so as to optimize the following objective function (Appendix A of Haarnoja et al. [7]):

$$J^{\text{static}}(\pi(a|s)) \equiv \sum_t \mathbb{E}_{(s_t,a_t) \sim \rho_\pi} \Big[ \sum_{l=0}^{\infty} \gamma^l \mathbb{E}_{(s_{l+t},a_{l+t}) \sim \pi} \big[ r(s_{l+t}, a_{l+t}) + \mathcal{H}(\pi(\cdot|s_{l+t}))|s_t, a_t \big] \Big] \tag{S11}$$

where $\rho_\pi$ is the probability of arriving at state $s_t$ at time $t$ and taking action $a_t$ when acting according to policy $\pi$. $\gamma$ is the usual discount factor, $r(\cdot, \cdot)$ is reward function and $\mathcal{H}(\pi(\cdot|s_t))$ is the entropy of policy $\pi$ in state $s_t$. The above objective corresponds to maximizing the discounted sum of rewards along with the entropy of the policy starting from every state-action pair $(s_t, a_t)$ (weighted with the respective probabilities).

In case of DIRL, we require our policy to be dynamic and vary with time. The policy given in Eq. 3 optimizes, at each timestep $t$, the objective function given by:

$$J^{\text{DIRL}}(\pi_t(a|s)) \equiv \sum_t \mathbb{E}_{(s_t,a_t) \sim \rho_{\pi_t}} \Big[ \sum_{l=0}^{T-t} \gamma^l \mathbb{E}_{(s_{l+t},a_{l+t}) \sim \pi_t} \big[ r_t(s_{l+t}, a_{l+t}) + \mathcal{H}(\pi_t(\cdot|s_{l+t}))|s_t, a_t \big] \Big] \tag{S12}$$

where $\pi_t$ is the time-varying policy and $r_t(\cdot, \cdot)$ is time-varying reward function. Here, we also switch to the finite horizon case, and the sum now goes to the maximum length of a mouse's trajectory, $T$. The proof that Eq. 3 optimizes this objective is a straightforward extension of that given in Appendix A of Haarnoja et al. [7].

Note: one limitation of this current objective is that it does not acknowledge the time-varying nature of the policy or reward function, and depends only on $\pi_t$ and $r_t$. As an alternative to maximizing Eq. S12 at each timestep, we could instead optimize the following objective, which is a function of the policy at all timesteps $\{\pi_t(a|s)\}_{t=1}^{T}$:

$$J^{\text{Modified}}(\{\pi_t(a|s)\}_{t=1}^{T}) \equiv \sum_t \mathbb{E}_{(s_t,a_t) \sim \rho_{\pi_t}} \Big[ \sum_{l=0}^{T-t} \gamma^l \mathbb{E}_{(s_{l+t},a_{l+t}) \sim \pi_{l+t}} \big[ r_{l+t}(s_{l+t}, a_{l+t}) + \mathcal{H}(\pi_{l+t}(\cdot|s_{l+t}))|s_t, a_t \big] \Big]$$
$$\tag{S13}$$

We chose to go with the approach of optimizing the DIRL objective given in Eq. S12 at each timestep for the following reasons:

1. It's not obvious which objective is more biologically plausible. It is unclear if we should assume that mice have full knowledge of their reward function for all future timesteps (Eq. S13) or if they are more likely to approximate their future reward function with their reward function at the current timestep (Eq. S12).

2. It is more computationally efficient to compute Q-values for the objective of Eq. S12. If we were trying to optimize the objective in Equation S13, we would likely have to do something akin to the backward pass in the forward-backward algorithm [8], and compute Q-values serially. By contrast, in the current set up, we can compute Q-values in parallel across timesteps.

Despite these reasons for working with Eq. S12, we think it would be a valuable research problem to investigate the optimal set of policies for Eq. S13.

# D    Additional DIRL results

## D.1    Simulations in the labyrinth environment

In Figure 3, we show the results of applying DIRL to trajectories simulated from a time-varying reward function in a $5 \times 5$ gridworld environment. While we focused on the gridworld environment there both to demonstrate the versatility of our approach, and to examine the performance of our framework in a standard IRL environment (IRL papers such as [9–11] all evaluate performance in a gridworld), it is important to also simulate within the labyrinth environment corresponding to the real world data shown in Figures 4 and 5. In Figure S2, we show the results of simulating 200 trajectories in the 127-node labyrinth using the reward function shown in panel D, a linear combination of the goal maps shown in panel A and the time-varying weights shown in panel B. As for the real data, we held out 20% of trajectories for model comparison purposes. Reassuringly, model performance peaked at the generative number of maps: 2. We

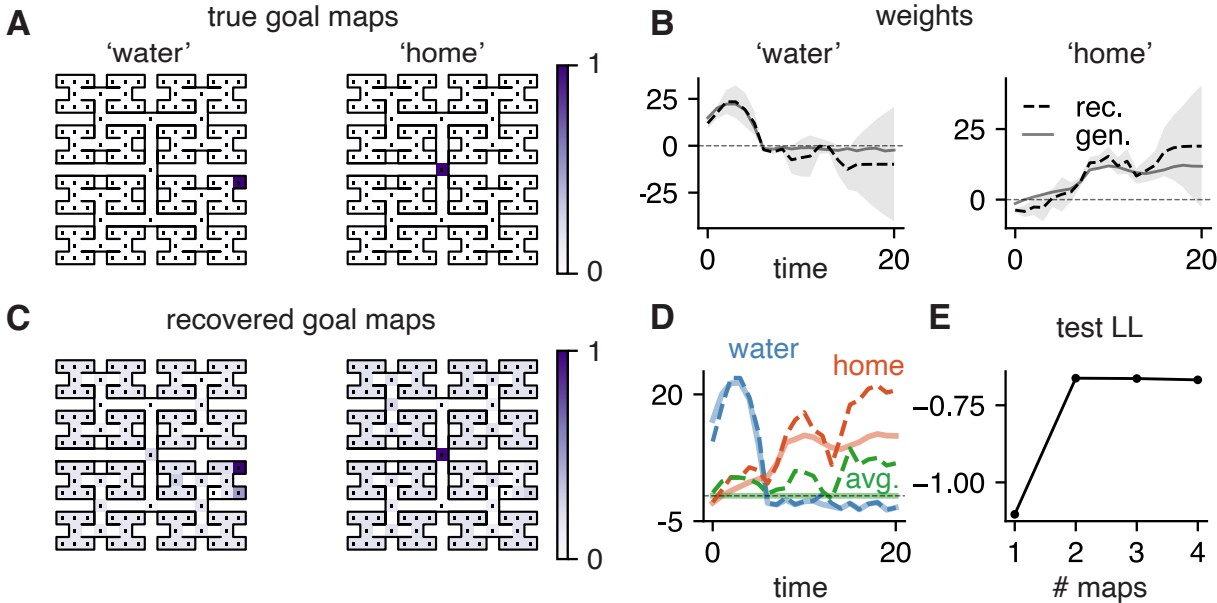

Figure S2: Simulations in the labyrinth environment. **(A)** Generative goal maps for labyrinth simulations: one goal map has the water state as highly rewarding, while the second has the home state as highly rewarding. **(B)** Generative (solid) and recovered (dashed) time-varying weights for 2-map solution. 95% confidence intervals, obtained via the inverse Hessian of the log-posterior (Eq. 6) at the MAP estimate of the weights, are also shown. Note: we use the same time-varying weights as were recovered from applying DIRL to the water-restricted mice in Figure 4. **(C)** Recovered goal maps in the 2-map solution. **(D)** Generative (solid) and recovered (dashed) time-varying reward function. We show the reward function for the water state (blue), the home state (red) and the average reward for all other states (green). **(E)** Validation set log-likelihood (in units of bits/decision) as a function of the number of goal maps. Reassuringly, this peaks at the generative setting of 2 maps.

show the recovered time-varying weights, goal maps and reward function for the 2-map solution in panels B, C and D respectively.

When performing this analysis, we swept across a similar grid of hyperparameters to those described in section A.3, which we document in Table S3 for completeness. As described earlier, we used validation-set performance to distinguish between different hyperparameter settings. Figure S2 corresponds to setting $K = 2$, $\sigma = 1$, $\lambda = 0.001$ and the learning rate for the maps to 0.05 and the learning rate for the weights to 0.01.

| Hyperparameter | Range |
|---|---|
| no. of maps, $K$ | $\{1, 2, 3, 4\}$ |
| noise variance, $\sigma$ | $\{1, 2^{-1}, 2^{-2}, 2^{-3}\}$ |
| $l_2$ coefficient, $\lambda$ | $\{1e-2, 1e-1, 1.0\}$ |
| learning rate for maps | $\{0.05, 0.01, 0.005, 0.001\}$ |
| learning rate for weights | $\{0.05, 0.01, 0.005\}$ |

Table S3: Hyperparameters for fitting DIRL on simulated labyrinth trajectories

## D.2 Additional results on real data: 3-map fits for water-restricted mice

In Figure 4 we focused on the 2-map fits for the water-restricted animals. However, since the 3-map solution obtained comparable validation performance for this cohort, we show the 3-map fits for this cohort here. As can be observed in Figure S3, the 'water' goal map is effectively repeated (maps 1 and 3 in panel A), so that the overall reward function (panel C) for the 3-map solution is similar to that obtained in the 2-map solution. Due to the fact that one map is effectively repeated – thus suggesting two principal maps – we chose to focus on the 2-map fits in the main text.

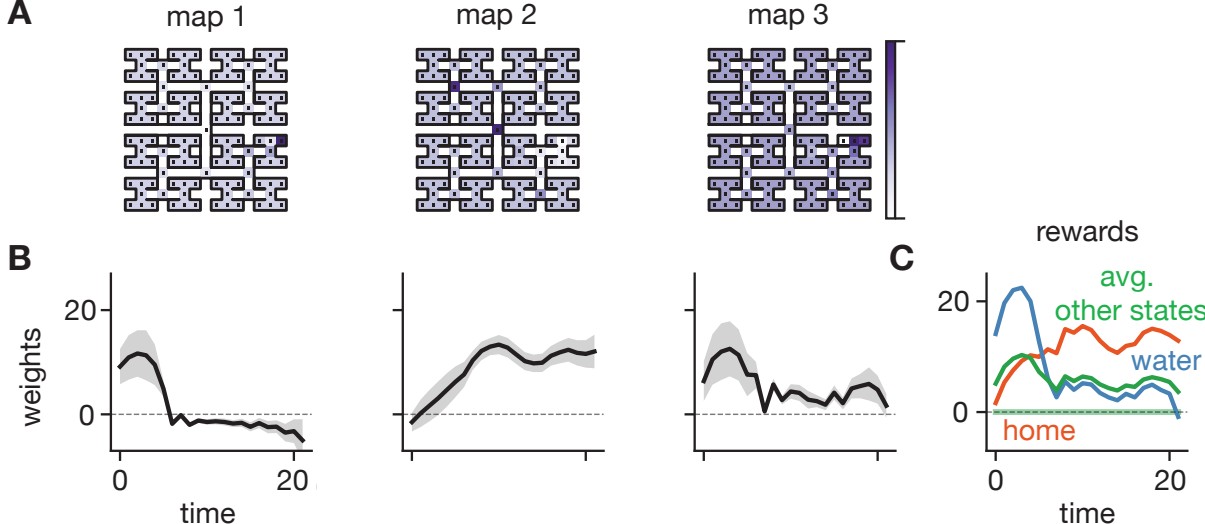

Figure S3: 3-map solution for water-restricted animals. **(A)** Recovered goal maps. Maps 1 and 3 are very similar. **(B)** Recovered time-varying weights placed on each of the 3 maps. **(C)** Recovered time-varying reward function: blue shows the reward for the water state, red shows the reward for the home state, while green shows the average reward for all other states.

### D.3 Trajectories simulated by DIRL better resemble mouse data compared to those from other IRL methods

In order to further demonstrate the utility of DIRL compared to existing IRL methods, we simulated trajectories using DIRL, maximum entropy IRL [1] and deep maximum entropy IRL [2] when parameters of these frameworks were fit to the trajectories of the water-restricted mice. In panels A-C of Figure S4, we show example simulated trajectories for the 3 methods. In panel D, we show the number of simulated trajectories (out of 200) that made it to the water state, and compare this to the number of real trajectories that made it to the water state in the same time period (199 out of 200). Surprisingly, *none* of the trajectories simulated by the maximum entropy and deep maximum entropy methods reached the water state in the time available, while 191 out of 200 reached the water port for DIRL. Clearly, DIRL is better suited to explaining the trajectory data of real mice compared to existing IRL methods.

## E  Code implementation of DIRL

At https://github.com/97aditi/dynamic_irl, we provide an implementation of DIRL. We use PyTorch 1.11.0 [12] to perform the optimizations of Equations 5 and 6. We also used code from the original Rosenberg et al. [4] paper (available at https://github.com/markusmeister/Rosenberg-2021-Repository) in order to produce some of our plots – our repository contains copies of the scripts we used.

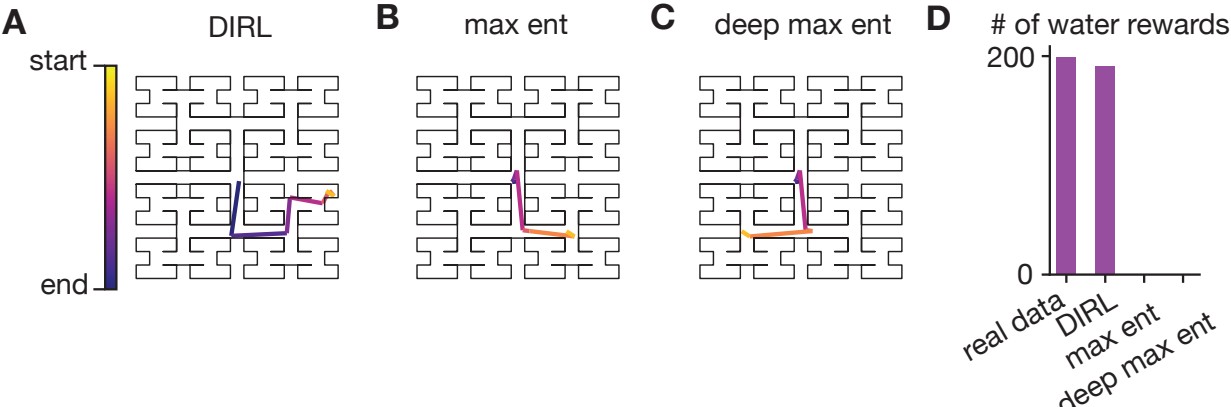

Figure S4: Simulated trajectories of length 8 for three different IRL methods (with parameters fit on the trajectories of water-restricted mice): **(A)** DIRL. **(B)** Maximum entropy framework of [1]. **(C)** Deep maximum entropy framework of Wulfmeier et al. [2]. **(D)** Number of water rewards obtained by each method in a total of 200 simulated trajectories of length 8 each (the maximum number of steps to reach the water port for the water-restricted mice is 8). Note that *none* of the trajectories for the Max. Ent. and Deep Max. Ent. methods made it to the water port, while 191 trajectories simulated by DIRL made it there.