# OpenReview forum: "Dynamic Inverse Reinforcement Learning for Characterizing Animal Behavior"
_NeurIPS.cc/2022/Conference — NeurIPS 2022 Accept_

### Official Review · Reviewer_PZrL · 2022-07-03

**Rating:** 10
**Confidence:** 5
**Soundness:** 4 excellent
**Presentation:** 4 excellent
**Contribution:** 4 excellent

**Summary:**

This paper describes an application of inverse reinforcement learning to maze exploration by mice. It proposes a novel IRL method to model changes in the reward function over time, by representing the reward as a linear combination of multiple goals, such that the importance of each goal can change over time. This model jointly infers the possible goals, and the dynamic reward function from mouse trajectory. The model can be applied to free exploration, where mice are not given an explicit goal on each trial. In this case it still recovers reasonable goal maps, such as a "home reward" and a random-looking exploration reward map. The model assumes that mice have multiple goals and choose between the set of actions associated with these goals using a Boltzman policy, but makes no other assumptions.

**Questions:**

1. Why is there a log in equation (4)? I am not familiar with a formulation of a Q function with a log.

2. Line 141: "coordinate ascent", line 144: "gradient ascent" -- this should have probably been "coordinate descent"?



**Limitations:**

I believe this work has minimal negative societal impact -- it is extremely unlikely that IRL can be used to infer human goals. The main limitation of IRL as acknowledged by the authors, is the need to see many instances of a behaviour. This will work well in contexts where the goal set is relatively small and constant through the observation period. However, IRL can easily miss fine-grained variation in goals, such as when animals see exploration as play,  trying out multiple ways to achieve a goal. This will show up as random actions when averaged over trials.

**Strengths And Weaknesses:**

Strengths: This is very interesting work -- among the most interesting NIPS submissions I have seen over the years, and a great application of IRL to a real problem. I strongly recommend this work to be presented at the conference. Although the explore map tells us little in practice, it is very interesting in theory and takes an important step toward understanding exploration. This is a short review because I think the paper is really great.

Weaknesses: the exploration map leaves some open questions  --
can this map emerge from a random walk?
from self-avoiding random walk?
are mice intentionally seeking out intersections, or other structural elements that allow them to encode the spatial map?
This is a minor weakness, since it is impossible to address all of these within 9 pages.

---

> ### Author Response · Authors · 2022-08-02
> **Response to Reviewer PZrL**
>
> We are very grateful to the reviewer for their extremely positive assessment of our work, and are glad that they found it interesting. We thank them too for their excellent suggestions for both extending our work so as to be able to better interpret the results returned by our model.  Finally, we thank them for their encouragement, and for recommending that our paper be presented at NeurIPS. Here are brief responses to their questions and comments:
>
> 1. *Exploration map*: We are grateful to the reviewer for their excellent suggestions for making sense of the retrieved ‘explore’ map. It would, indeed, be exciting to simulate trajectories that emerge from a random walk, a self-avoiding random walk, and from agents that have preferences for particular structural aspects of the maze environments we consider. While this paper is predominantly focused on methodology, we’re very interested in pursuing a separate follow-up paper discussing the neuroscientific implications of DIRL, and the meaning of the ‘explore’ goal map. We would love to incorporate those great suggestions there.
> 2. *Log in the Q-function*: It is true that standard value iteration does not have a log in the Bellman equation. However, here we use soft value iteration, which has been used over the years in the RL community for several reasons including differentiability of the learned policy (see, for example, Harnooja et al., 2017; reference [37] in our paper). Inference in our method requires the policy to be differentiable, hence we reverted to soft value iteration.
> 3. *Coordinate ascent vs descent*: We are doing coordinate ascent here as we aim to maximize the log posterior of the observed trajectories under our model (i.e. the probability of generating the observed trajectories under our model).

---

### Official Review · Reviewer_Rzrz · 2022-07-10

**Rating:** 6
**Confidence:** 3
**Soundness:** 3 good
**Presentation:** 4 excellent
**Contribution:** 3 good

**Summary:**

The paper proposed a new method, Dynamic IRL, and applied it to model mouse behaviour in a spatial navigation task to recover the goals of the animals. Unlike the previous IRL methods, the Dynamic IRL allows the reward to change over time slowly. To make the inference trackable, the authors model the reward function as a linear combination of weights and a few "goal maps". The authors demonstrated and validated the model on a simulated gridworld task, and apply it to a real mouse dataset to successfully recover the goals of the animals in two different conditions.

**Questions:**

"Building computational models of decision-making is a core objective in both neuroscience and psychology" - this is a substantial statement and I believe it is not 100% true. It might be true for some computational neuroscientists/psychologists, but the goal of building models is not to build models, it is for understanding the biological system or psychological system. I would suggest authors make this softer, especially when this is the first sentence of the abstract and introduction.

From the paper, it seems that the author separate the data into 80% training data, and 20% testing data (where test LL was computed). When tuning the hyperparameters, the authors did it on a validation set. Did the author use a separate validation set from the 80% training set, not using the 20% testing data? It would be great if the author can make this very clear in the writing.

Following the training data split, I'm wondering if the non-distinguishable 2-map and 3-map can be due to a lack of statistical power. For example, in the simulation, the authors can try using longer sequences or more sequences, just to test this hypothesis (even though longer or more sequences are not available in the real dataset). Or instead of an 80/20 split, k-fold cross-validation might provide more statistical power.

If this non-distinguishable is intrinsic not from model recovery noise, I would be happy to see (brief) discussion on future work on addressing this, maybe somehow penalize for model complexity in the model?

The data selection part can seem a bit arbitrary. It makes sense to remove the first 25 or so trajectories, but I'm wondering what else (if any pattern) is removed. I'm also wondering if the results change dramatically if this data selection is not done (maybe only removing the first 25), i.e. is this step necessary when apply the dynamic IRL model? If so, I would expect the model can capture some noise to some extent, e.g. the reason to allow reward function to vary is to capture fatigue. I would like to see a (brief) discussion on future work on a more principal way to capture noise. Maybe a noise map, or allowing the reward function to not just change slowly across time, but also can undergo change point?

It is exciting to see the model is already applied to a real animal dataset. However, the conclusion/implication we can draw from the modelling results seems to only confirm what we can observe from the behavioural data alone, and mostly serve to validate the model's ability to recover something meaningful. I can imagine the recovered weights can be used to correlate with some neural recordings. The authors touched on this a bit by saying the method can provide a normative account, but since this is mainly a methodology paper, it would be great to add a sentence or two, detailing what kind of scientific findings your model can provide in addition to only analysing behavioural data.







**Limitations:**

Another limitation might be, that in lots of behavioural experiments, the state might be hidden/unobservable. I think this is definitely out of the scoop of this paper, that the authors do not need to include/address. But if authors have comments on this, it can be very interesting to read.

**Strengths And Weaknesses:**

Strengths:
- The paper is well-written and easy to follow. The figures are also high quality.
- The contribution is clear, allowing the reward to vary across time in the IRL model.
- The authors performed experiments on both synthetic data and real animal datasets, and compared to an existing IRL model.

Weaknesses:
(I only list them here, and I will explain more and put my questions and suggestions in the next section)
- The number of goal maps seems to be non-distinguishable to some extent, e.g. in the simulation, the 2-map and 3-map fit the 2-map data equally good.
- When applying it to real data, it requires a procedure to cluster the data first.
- The modelling results seem to mainly confirm the hypothesis from the behaviour and descriptive.

---

> ### Author Response · Authors · 2022-08-02
> **Response to Reviewer Rzrz**
>
> We are delighted that the reviewer found our paper to be “well-written”, “easy to follow”, to have “high quality” figures and that they found “the contribution [to be] clear”.
>
> We will now respond to each question/suggestion for improvement in turn:
> 1. *Refactor "Building computational models of decision-making is a core objective in both neuroscience and psychology" sentence.* Thank you for this suggestion. We agree, and will alter the opening sentence to: “Understanding decision-making is a core objective in both neuroscience and psychology, and computational models have often been helpful in the pursuit of this goal.” We welcome suggestions for further improvement.
> 2. *Dataset for selecting hyperparameters*: We apologize for not making this clearer in the paper. Due to the relatively small number of trajectories that we study, we use the same 20% of trajectories for selecting hyperparameters and for reporting validation performance. While not ideal, it is not uncommon to do this in applications involving biological data (where datasets are small and expensive to collect): for other examples of this see Dezfouli et al. (2019) or Ashwood et al. (2022). We will aim to clarify this point in all future iterations of our paper.
> 3. *Non-distinguishable 2- and 3-map fits*: The non-distinguishability of the 2- and 3-map fits is a consequence of the nesting of models that occurs as the number of goal maps in the model increases. That is, the 2-map model corresponds to a 3-map model where the time-varying weights for the third map are zero.  Alternatively, the 2-map model corresponds to a 3-map model where one of the two maps is repeated and the time-varying weights for the repeat map are split so as to sum to the overall generative weights.  We see the latter scenario in Figure S3 where we show the 3-map model fit for the water-restricted mice, and see that the water goal map is recovered twice, while the time-varying weights for the water maps sum to give the same time-varying weights for the water map as in Figure 4.  We do not view this as a weakness of our approach: such nesting is not unique to our framework, and any mixture model or Hidden Markov Model has this property. However, the reviewer is correct that this is something we have to take into account when performing model comparison, and we select the number of goal maps by observing where validation performance, as a function of the number of goal maps, begins to level off. We will seek to make this point clearer in the final version of the paper.
> 4. *DIRL with hidden/unobservable states:* That’s a good point. One way to address the fact that states are not fully observable in some settings is to cast the problem as a POMDP, and in fact, Reviewer yX2q has pointed us to a paper that applies IRL with POMDPs. It may be possible to augment DIRL with hidden/unobservable states, however, as the reviewer acknowledges this is out of the scope of this paper. We will discuss this in the future work section.
> 5. *Data selection procedure/ability of DIRL to capture noise*: We agree that it would be more elegant to fit DIRL to all behavioral data, learn goal maps from all decisions together, and then a separate set of time-varying weights for each trajectory. Future work could involve developing a hierarchical model for the time-varying weights, where the weights for an individual trajectory are constrained to be ‘close’ to those for other trajectories. If we were to employ this hierarchical model for the time-varying weights, and then learn goal maps from all decisions concatenated together, this is likely to reduce the effect of noise on the retrieved goal maps and time-varying weights, assuming that noise in the generative maps and weights is uncorrelated. While beyond the scope of the current paper, we would be happy to clarify the reasoning behind our current data selection procedure, as well as our proposed extension in text.
> 5. *Scientific findings offered by DIRL*: please see response to all reviewers.

---

### Official Review · Reviewer_yX2q · 2022-07-11

**Rating:** 6
**Confidence:** 2
**Soundness:** 3 good
**Presentation:** 3 good
**Contribution:** 3 good

**Summary:**

# Summary
This paper presents an inverse reinforcement learning algorithm designed specifically for navigation tasks involving non-stationary reward functions.
The approach is motivated by applications in neuroscience, for example understanding the (time-varying) goal-seeking or exploratory behavior of animals.
The algorithm builds on maximum entropy IRL, it assumes access to the transition model of the environment (in performing soft value iteration), and is designed for discrete state spaces (in how it defines features).
The rewards are defined as a linear function of learned goal-map features and learned weights associated with those features.
The non-stationarity of the environment rewards is handled through learning feature weights at each timestep.
The approach is evaluated  in a simulated domain matching the assumptions of the algorithm as well as on real-world data collected from (water restricted and unrestricted) mice navigating a maze.

# Contribution
- I believe the contribution is a method (based on maximum causal entropy IRL) for inferring (time-varying) reward functions of animals performing navigation tasks, specifically in settings with discrete states, deterministic transitions, and where time-alignment of trajectories is practical


**Questions:**

The section above includes a number of questions. In addition to those:
1. What are the convergence properties of the algorithm (specifically the dual ascent of the features / weights)? If you run it multiple times, how much do the learned features differ?

And some minor questions:
1. Section 3.1 introduces MDPs, but it’s not clear to me whether the paper deals with finite horizon MDPs (as would be indicated by the use of a timestep-dependent reward function and Q-function) or infinite horizon MDPs (as would be indicated by the use of a discount factor)
2. In equation 4, should the q-function on the rhs have time index t + 1?


**Limitations:**

Thank you for including a section discussing the limitations of the approach


**Strengths And Weaknesses:**

# Strengths
1. The paper is clear and well-written
2. The paper is motivated by an important problem (better understanding human/animal behavior)
3. The paper focuses on less common aspects and applications of IRL. For example, the application of IRL in a neuroscience context is interesting and less common than applications in other areas (e.g., robotics or autonomous driving), and applying IRL in a context where learning rewards is itself a primary goal is interesting (for example, the differences in the learned goal maps between the two mice cohorts).
4. The approach does seem applicable to understanding animal behavior in situations similar to those considered in the paper
5. The supplementary material provides extensive details on the algorithm and experiments (including well-documented code allowing for reproducing experiments), and execution of the experiments seems sound


# Weaknesses
## Originality
- The main algorithmic novelty seems to be learning time-varying rewards; however, it seems plausible to me that this non-stationary aspect could instead be addressed by treating the problem as partially observable (where the timestep or the history of the agent are unobserved).
- In the context of _RL_ non-stationary MDPs can be formulated as _RL_ within a specific type of partially observable MDP [4]. Does that connection hold for _IRL_? For example, can you simply specify the timestep as a feature (or specify a feature capturing historical information) and thereby remove the need for time-varying rewards? Can you instead use IRL methods designed for partially-observed environments [5]? What's the benefit of the time-varying reward formulation over these alternatives?
- Separately, what’s the relationship with existing work that discusses IRL in the context of non-stationary reward functions such as [3]?

## Relationship with prior work
- The paper would be improved by clarifying the relationship between the proposed algorithm and existing work (citations to relevant work are provided in section 3.2, but describing the relationship specifically would help)
- The algorithm seems to be a special case of maximum causal entropy IRL [1]. Both algorithms alternate between the two steps of (i) computing a policy on the current set of rewards via soft VI and (ii) updating rewards to maximize the likelihood of the observed data under the learned policy. This approach is discussed in [1], and a more recent discussion that makes the connection clear is given in [2] (section 3.2). In particular, the proposed algorithm assumes a deterministic transition function, and therefore employs simplified likelihood calculations in eqs (5) and (6) as discussed in section 3.3 of [2].
- The algorithm learns parameters of both the features and their weights, which existing work does as well (e.g., reference 30 in the paper), though with a particular structure and using coordinate ascent.
- Additionally, it would help to distinguish from IRL work in understanding human behavior. For example, what’s the relationship with IRL methods used for predicting pedestrian behavior in navigation tasks, for example [6,7]?

## Significance
- The approach doesn't seem to scale to large / continuous state/action spaces (due to the use of soft value iteration and defining features per-state). Section 5 discusses an extension that would learn embeddings, but it’s not clear to me how this would work.
- The approach doesn't seem applicable in cases where the transition model is stochastic and/or unknown (due to using the deterministic transition model form of the likelihood in eq (5) and (6), and due to using soft VI).
- The structure of the features seem specific to discrete state-space navigation tasks
- Time-varying rewards seem difficult to apply in many situations. For example, the clustering and alignment procedure used in the paper (that seems to require each trajectory to be aligned and of the same length) would be difficult to apply to trajectories of animals/humans in the real world (where natural start and end points are likely unavailable).
- Despite the above limitations, it does seem the approach is applicable to understanding animal behavior in situations similar to those considered in the paper. I think this is significant, but it's not clear to me how significant. I think the paper would benefit from further arguing why this particular setting is important.

## Experimental quality / analysis
- Why do MaxEntIRL and DeepMaxEntIRL perform poorly? Section 4.2.3 claims they perform poorly because they learn a static reward function, but is it possible the issue can instead be framed as one of partial observability?
- For example, if these baselines were provided with features capturing the timestep or history, how would they perform? To capture the history a map of normalized state-visitation counts could be provided as a feature.
- If addressing partial observability (of the timestep or state-history) addresses the issue, why should we prefer learning time-varying rewards?
- Another way to potentially answer this question would be to train a (recurrent) policy via behavior cloning, and then compute likelihood on the validation set.
- Providing an additional experiment on a different real-world dataset would improve the paper by illustrating the generality of the approach


[1] Modeling Interaction via the Principle of Maximum Causal Entropy

[2] A Primer on Maximum Causal Entropy Inverse Reinforcement Learning

[3] Dealing with multiple experts and non-stationarity in inverse reinforcement learning: an application to real-life problems

[4] Hidden-Mode Markov Decision Processes for Nonstationary Sequential Decision Making

[5] Inverse Reinforcement Learning in Partially Observable Environments

[6] Intent-aware long-term prediction of pedestrian motion

[7] Learning How Pedestrians Navigate: A Deep Inverse Reinforcement Learning Approach

---

> ### Author Response · Authors · 2022-08-02
> **Response to Reviewer yX2q [3/3]**
>
> **Experimental quality/analysis**
> 1. *Performance of Maximum Entropy and Deep Maximum Entropy frameworks relative to DIRL*: Thank you for the suggestion of augmenting the baseline models considered in our paper to include features capturing timestep or history. It is certainly possible to extend the definition of ‘state’ in the Maximum Entropy and Deep Maximum Entropy frameworks to use not just the physical node in the environment, but also a map of node-visitation counts, although this has the effect of increasing the size of the state space dramatically. In particular, so that the transitions of the MDP continue to be Markovian, we need to keep track of node-visitation counts for all nodes in the environment. If we simplify the problem and instead just keep track of whether or not each node has been visited, even this increases the size of the state space from $127$ to $127 \times 2^{127} \sim 10^{40}$!   Thus, it isn’t straightforward to apply the Maximum Entropy and Deep Maximum Entropy frameworks with node-visitation counts.  However, to check whether or not these baseline frameworks could perform better when given access to features capturing timestep or history, we performed an experiment where we augmented the state space for these frameworks by conditioning on whether or not the animal’s previous action led to it moving deeper into the labyrinth (further from the home port). This effectively allows us to learn a distinct reward map for early and late stages of an animal’s trajectory: in earlier timesteps, the animal often moves from the home port to locations deeper in the labyrinth, while at later timesteps, it will either stay at the same location or moves toward the exit. We found that endowing the maximum entropy and deep maximum entropy frameworks in this way did result in a performance improvement: using the rewarded animals dataset as an example, test loglikelihood increased from -1.76 bits/decision to -1.51 bits/decision for the maximum entropy framework and from -1.71 bits/decision to -1.35 bits/decision for the deep maximum entropy framework.  However, these improvements were still not large enough for these frameworks to outperform DIRL (our method; test loglikelihood for 2 map solution was -0.83 bits/decision). While conditioning on additional past actions may improve the performance of these frameworks further, the state space of the MDP blows up dramatically as the number of past actions increases, putting a limit on the total number of past actions that can be considered.
> 2. *Compare DIRL to a behavioral cloning method*: While it may be useful to fit a behavioral cloning policy to the animal’s data in order to obtain an upper bound on the predictability of the animal’s decisions, a black box policy is not, in itself, very useful for the neuroscientific settings we consider.  In particular, we also want interpretable explanations of mouse behavior that can be readily correlated with neural activity. See our general response for a more detailed discussion of this point.
>
> **Other questions**
> 1. *Convergence properties*: As with any dual coordinate ascent algorithm, our inference procedure can converge to multiple local optima. To avoid this, we used 10 initializations during inference and selected the result with the highest validation log-likelihood in the paper (we verified in simulation that 10 initializations was enough to reliably avoid local optima).
> 2. *Finite vs infinite horizon MDP*: Thank you for the comment, we will clarify this in text. In particular, we consider a finite horizon MDP, however, also include a discount factor for biological plausibility.  A rich literature exists within psychology and behavioral economics (e.g. Tversky and Kahneman, 1986), that suggests that humans employ temporal discounting when making decisions; we assume this to be true for animals too.

---

> ### Author Response · Authors · 2022-08-02
> **Response to Reviewer yX2q [2/3]**
>
> **Significance**
>
> 1. *DIRL doesn’t readily scale to large state/action spaces*: The reviewer is correct that, due to the use of soft value iteration and defining features per-state, DIRL does not currently scale well to large state/action spaces. However, we do not view this as a serious limitation of our approach given that in the neuroscientific decision-making experiments that we are interested in characterizing, the 127 node labyrinth of Rosenberg et al. (2021) is considered to be a large state space. Thus, our method already scales to sizes appropriate for the intended primary domain of application. In future work, we will certainly consider new innovations to improve scaling, such as the one described below.
> 2. *Clarification on proposed extension to DIRL*: We also wanted to clarify our proposed extension to DIRL that utilizes state embeddings, $\phi(s) \in \mathbb{R}^{D}$, rather than state identities. In particular, we would now learn goal maps in the embedding space (so that goal maps would lie in $\mathbb{R}^{\text{Num. Maps} \times D}$). The reward for state $s$ at time $t$ would be given by $r_{s,t} = \sum_{i}^{\text{Num. Maps}} \alpha_{i, t} u_{i}(\phi(s))$, and state embeddings, $\phi(s)$, could be the output of a neural network, as in Wulfmeier et al. (2015).  By moving to state embeddings, this would allow for generalization across states.  Furthermore, provided $D$ is small, this would also allow us to scale our approach to higher dimensional state spaces. Implementing this proposed extension is beyond the scope of the current paper (indeed, this could be a paper in itself!), as key details such as how to set the dimensionality of the embedding space, as well as the development of an inference method that allows for learning both state embeddings as well as the goal maps and time-varying weights will have to be worked through fully (for example, should the state embeddings be learned separately from the reward parameters, or can these be learned together end-to-end?).
> 2. *Transition matrix can’t be stochastic or unknown*: We would like to clarify that DIRL can be applied in cases where the transition matrix is stochastic: eq. 4 does not require a deterministic transition matrix, $P(s’|s, a)$. However, the reviewer is correct that DIRL does require that the transition matrix is known. We do not view this to be a major weakness of our approach as, after the initial 25 or so trajectories in the maze, we expect the mouse to have knowledge of the approximate structure of the environment. However, future work could investigate extending our approach to perform dynamic IRL with noisy models of the transition matrix of the environment.

---

> ### Author Response · Authors · 2022-08-02
> **Response to Reviewer yX2q [1/3]**
>
> We thank the reviewer for their detailed comments and analysis of our paper. We are pleased that they found our paper to be clear and motivated by an important problem. We are also glad that they found “the application of IRL in a neuroscience context [to be] interesting and less common than applications in other areas”. In general, we felt that the content and tone of the review were more positive than indicated by the reviewer’s score, so we humbly request that the reviewer consider raising their score if they would like this work to reach a broad audience.
>
> We will now respond to each question/suggestion for improvement in turn:
>
> **Originality/Relationship to other work**
> 1. *On Partially Observed MDPs vs time-varying rewards*: We apologize for not citing [4] and [5]: we will certainly include these in the related work section of our paper going forward. The reviewer is correct that if we assume that rewards switch between a known number of discrete states (and not continuously as in DIRL), then a similar problem to the one we study can be formulated as the HM-MDP introduced in [4]. Furthermore, [4] shows how to cast an HM-MDP as a POMDP, and [5] describes how to perform IRL for POMDPs under certain assumptions. However, we are not aware of any work that has fully investigated performing IRL with the HM-MDP structure (there would, undoubtedly, be small details that would need to be worked out in order to apply [5] to [4]), nor are we aware of any work that performs IRL with a continuously varying POMDP structure. In our setting, as we are modeling effects such as fatigue, satiety, and changes in intrinsic motivation, we believe that these can be better captured by a model with continuously-varying, rather than discrete, rewards. Finally, compared to methods utilizing POMDPs, our approach has the benefit of providing interpretable explanations of animal decision-making behavior (via goal maps and time-varying weights) that can be readily understood by non-RL experts.
> 2. *Clarify connection to prior works such as ref [30] in the paper, reviewer’s refs [1,2]*: Thank you for the comment. We will make sure to better distinguish our contributions from past work in sections 3.1 and 3.2. In Figures 4 and 5 we have compared our method to [30] (this is the "deep max ent" IRL method in our figures). Compared to [30], which learns a static reward and static state features, our method learns a time-varying reward function. However, the reviewer is correct that our inference method builds upon that in [30], as well as those used in the reviewer’s references [1] and [2]: we will make sure to fully acknowledge this going forward.
> 3. *Connections to existing IRL papers with non-stationary rewards*: The IRL algorithm with non-stationarity rewards in [3] assumes that the agent has a set of K discrete intentions with a discrete reward function corresponding to each intention. This is distinct from our setup where we expect the reward to vary on shorter timescales, as captured by a reward function that varies continuously over time. This allows for a more flexible representation of the behavior of mice, as the variables that we expect to be determining their reward functions are continuous in nature (satiety, fatigue, etc). We are certainly grateful for this reference and will add this to the related work section of our paper.
> 4. *Relationship of our approach to IRL for modeling human behavior*: Thank you for pointing us to these papers which we will discuss in the related works section of our paper. Here we briefly describe the differences between our approach and the cited papers:
>  [6] does not infer a reward function, which is the core aim of our paper. They assume that the reward is known. The goal of [6] is to learn a non-stationary policy arising from the dependence of the policy on a “goal” variable, which captures the intent of the pedestrian and is a discrete variable. Hence, a mixture of policies is being learned, each capturing a different goal. This is different from our setting where the policy varies over time as a consequence of the rewards varying continuously over time, while also learning rewards.
> While [7] indeed uses IRL to learn the rewards of pedestrians, these rewards are not time-varying but instead static. Furthermore, the goal of that paper is to incorporate the behavior of other pedestrians into the feature vector of the pedestrian being modeled. This is distinct from our application, where we are not modeling decision-making in the context of social behavior.

---

> ### Comment · Reviewer_yX2q · 2022-08-08
> **Response to author comments**
>
> Thank you for thoroughly responding to my comments. Your responses address a number of my concerns above, and I will increase my rating of the paper as a result.

---

### Official Review · Reviewer_9v8k · 2022-07-12

**Rating:** 8
**Confidence:** 5
**Soundness:** 4 excellent
**Presentation:** 4 excellent
**Contribution:** 3 good

**Summary:**

This paper introduces an inverse reinforcement learning framework that infers the time-varying reward function of the animal. The reward is designed as a linear combination of rewards maps, which can be weighted differently. The proposed method is confirmed using simulated data and real mouse trajectories.

**Questions:**

Overall, the reviewer enjoys reading this paper. It is excellent work, and the NeurIPS community will like it. However, there are a few clarification questions as follows.
- How is the artificial mouse trained in Section 4.1? The reviewer assumes that the authors train the artificial mouse to generate the trajectories. Is any specific RL algorithm used to train the agent?
- All the results included in the paper are based on two maps scenario. The inferred weights in Figure 4B and Figure 5B are only two types. Three maps scenario is presented in the Supplementary material. However, Is it possible to extend this method to multiple maps? Maybe more than 5? If then, this work can be useful in the ML and neuroscience communities.
-  Some essential pieces of literature are missing. There are recent NeurIPS papers that study RL or IRL in decision-making problems in neuroscience. These can be discussed in related works.
 "A neurally plausible model learns successor representations in partially observable environments." NeurIPS 2019,
 "Inverse rational control with partially observable continuous nonlinear dynamics." NeurIPS 2020.




**Limitations:**

The authors describe the limitation in the last section.

**Strengths And Weaknesses:**

Strength
-      Interesting story with important motivation. The inverse RL approach to time-varying reward is novel.
-      Well-written paper. Easy to follow. All figures are easy to read and well-organized.
-      Supplementary material is helpful and well-organized.

Weakness
-      Little impact on the ML community (in terms of ML algorithm). Nevertheless, it is a valuable work to the neuroscience community.
-      Labels are missing in some figures, which makes it hard to interpret the graph.

---

> ### Author Response · Authors · 2022-08-02
> **Response to Reviewer 9v8k**
>
> We are delighted that the reviewer found our work to be interesting and important. We are thankful for their comments and suggestions. Here are our responses to the questions/weaknesses raised by the reviewer:
>
> 1. *Missing labels in graphs*: We often used the title of a panel in a figure to explain what was being conveyed by the y-axis, and we also avoided repeating axis labels for e.g. time-varying weights corresponding to different goal maps. These were stylistic choices designed to reduce the overall amount of text in a figure; however we apologize for any confusion caused by these choices and will do our best to clarify them. We also welcome suggestions the reviewer might have for making our plots clearer.
> 2. *Training of the artificial mouse in section 4.1*: We first generated the smooth time-varying rewards shown in Fig 3E (solid line) by combining the two goal maps shown in Fig 3B, with the time-varying weights shown in Fig 3C (grey solid line) according to equation 1. Using these known rewards, we computed the simulated animal’s optimal time-varying policy for this reward function (eq. 4) using soft value iteration (eq. 3). We then generated trajectories by selecting a start location randomly before executing the retrieved time-varying policy for 50 time steps. We will ensure that we clarify the trajectory generation process in the revised manuscript.
> 3. *Extension to multiple maps*: Our method is certainly not limited to two or three maps, and we apologize for the confusion on this point. In fact, in Figures 4F and 5F, we show the test set performance of DIRL with 4 maps. In these experiments, we did not go beyond 4 maps as the test log-likelihood started saturating at 2 maps, but our method itself is not restricted to <5 maps. We also agree with the reviewer that the ability to capture a larger number of maps will increase the large-scale utility and applicability of our method.
> 4. *Relevant literature*: We are grateful for these references, and we will certainly add them to our related work section.

---

### Author Response · Authors · 2022-08-02
**Response to all reviewers**

We thank the reviewers for their enthusiastic assessment of our work, and for their detailed comments and helpful suggestions. We were delighted that 3 out of 4 reviewers placed our paper above the acceptance threshold, and that reviewer PZrL found our paper to be “among the most interesting… submissions I have seen over the years”. We were pleased that reviewer 9v8k found our paper to be an “interesting story with important motivation”.  We thank Reviewers PZrL and 9v8k for their confident and positive assessments of the paper, and humbly request that Reviewers yX2q and Rzrz consider raising their scores if they would like this work to reach the broader NeurIPS community.

We will first discuss some general points raised by multiple reviewers, and then address reviewer-specific comments.

**Scientific findings offered by DIRL (yX2q, Rzrz)**: One novel insight that DIRL provides us, which was not obvious from the data alone, was the exploration tendencies of the water-unrestricted mice. As acknowledged by reviewer PZrL, this is an “important step towards understanding exploration” in animals, and future work can seek to provide a normative account for the nature of the exploration map. Moreover, we anticipate DIRL being useful in correlating the behavior of mice with their neural activity: it will be interesting to study how neural activity varies with fluctuations in the animal’s time-varying goals.

**Additional datasets where DIRL could be applied (yX2q, Rzrz):** While we developed DIRL with neuroscientific applications in mind (e.g. characterizing animal decision-making during spatial navigation or foraging), our framework is general and can readily be applied in other fields. For example, one could imagine applying DIRL to healthcare settings, such as those considered in  Hüyük et al. (2022). In Hüyük et al. (2022), the authors acknowledge that the policies used by healthcare providers to allocate organ donations have varied over time. By applying DIRL to a similar dataset to the organ allocation dataset studied in that paper, it may be possible to recover the latent goals of policy-makers or medical practitioners when deciding how best to allocate organs to patients.

Reference:
Hüyük, Jarrett and van der Schaar (2022), *Inverse Contextual Bandits: Learning How Behavior Evolves over Time*

---

### Meta-Review · Area_Chair_BNz3 · 2022-08-27

**Recommendation:** Accept
**Confidence:** Less certain

**Metareview:**

Unanimous acceptance recommendations.

Reviewer yX2q had a long list of weaknesses, and was the most thorough reviewer, but most have since been addressed by the authors. While prior time-varying IRL prior methods exist, their application their dynamic inverse reinforcement learning (DIRL) method to animal decision-making in more complex tasks from to real animal (mouse) data seems uncommon yet relevant to NeurIPS. There are some concerns about scalability, but for these applications like inferring mouse goals and rewards in maze like environments, scaling isn't as necessary.

From my own reading of this work, I agree with the reviewers it's also interesting, and worth accepting. The authors' modelling of the low-rank time-varying reward structure was nice and non-trivial, and it does give interesting interpretability to a discrete number of mice behaviors it discovers like "explore", "go home", "find water". So the application and method seem interesting and novel. It's possible this could be better suited to a neuroscience conference instead, but the use of IRL here might complicate that, so being between fields, the best venue for a work like this is possibly NeurIPS.

While I recommend accept here, the higher scoring reviewers were not overly specific in the merits of the method to warrant an award, so I interpret their scores as slightly overzealous, and so without significant algorithmic novelty beyond the time-varying modelling in this context (which was nice but not ground breaking), I'll not argue for an award.

**Award:**

No

---

### Decision · Program_Chairs · 2022-09-14

Accept